# Homocoupling Reactions of Azoles and Their Applications in Coordination Chemistry

**DOI:** 10.3390/molecules25245950

**Published:** 2020-12-15

**Authors:** Steffen B. Mogensen, Mercedes K. Taylor, Ji-Woong Lee

**Affiliations:** 1Department of Chemistry, University of Copenhagen, Universitetsparken 5, 2100 Copenhagen Ø, Denmark; lng656@alumni.ku.dk; 2Center for Integrated Nanotechnologies, Sandia National Laboratories, Albuquerque, NM 87185, USA

**Keywords:** homocoupling, bipyrazole, transition-metal catalysts, metal–organic frameworks

## Abstract

Pyrazole, a member of the structural class of azoles, exhibits molecular properties of interest in pharmaceuticals and materials chemistry, owing to the two adjacent nitrogen atoms in the five-membered ring system. The weakly basic nitrogen atoms of deprotonated pyrazoles have been applied in coordination chemistry, particularly to access coordination polymers and metal-organic frameworks, and homocoupling reactions can in principle provide facile access to bipyrazole ligands. In this context, we summarize recent advances in homocoupling reactions of pyrazoles and other types of azoles (imidazoles, triazoles and tetrazoles) to highlight the utility of homocoupling reactions in synthesizing symmetric bi-heteroaryl systems compared with traditional synthesis. Metal-free reactions and transition-metal catalyzed homocoupling reactions are discussed with reaction mechanisms in detail.

## 1. Introduction

Heterocyclic compounds containing two or more nitrogen atoms display unique molecular properties, which can be extended to macroscale properties in materials chemistry [1]. Furthermore, symmetric, multiply-functionalized organic ligands are required to construct ordered materials, preferably in high crystallinity [2]. Homocoupling reactions of aromatic compounds are often utilized to form carbon–carbon bonds in the presence of (over)stoichiometric or catalytic amounts of transition metal species [3]. For example, Ullmann coupling reactions can enable facile access to symmetric biaryl compounds using copper reagents with or without ligands at high reaction temperatures [4,5]. Based on this method, a plethora of reports was dedicated to the homocoupling of aromatic compounds, accessing symmetric molecules starting from simple aryl halides and derivatives. However, the reported procedures are often limited in terms of substrate scopes, particularly with heterocyclic compounds. Therefore, milder and more practical reaction conditions are required. To this end, this review summarizes recent development in (catalytic)homocoupling reactions of 5-membererd ring heterocycles, particularly focusing on pyrazoles due to their importance in materials chemistry as symmetric bipyrazoles. First, we will discuss traditional syntheses of bipyrazoles without homocoupling reaction methods, to highlight the advantages of (catalytic)homocoupling reactions. The following sub-chapters are organized by metal-catalyzed and metal-free reactions to illustrate the developed synthetic methods. The utility of bipyrazole materials in materials chemistry is discussed in Section 3.3, where we provide ample information on constructing metal–organic frameworks based on bipyrazole ligands.

## 2. Traditional Syntheses of Bipyrazoles

Condensation reactions of 1,4-dicarbonyl compounds with heteroatom nucleophiles can provide various 5-membered heterocycles: furans, thiophenes, pyrroles and their derivatives (Scheme 1A) [6,7]. To access pyrazoles with the same manner, 1,3-dicarbonyl compounds and hydrazine are required (Scheme 1B). However, the lack of a general method of “dimerization” of pyrazole substrates was a major bottleneck for the synthesis of symmetric bipyrazoles. Alternatively, diamino substrates can be prepared to access pyrazoles via oxidation reactions [8]. In 1964, Trofimenko reported the synthesis of 1,1,2,2-ethanetetracarbaldeyde, which was converted to bipyrazole by a condensation reaction with hydrazine (Scheme 1C) [9]. There are a few synthesis methods to access bipyrazoles via lengthy synthetic steps starting from diethyl succinate via a 3,4-carbonyl furan intermediate (**1**) [9,10,11].

Starting from succinic ester, a repeated formylation and acetal protection sequence afforded the 3,4-dicarboxylic furan intermediate **2** after treating with concentrated sulfuric acid in 66–68% yield (Scheme 2A). The reaction employed sodium metal and orthoformate for formylation reactions. The author reported that the reaction was amenable to characterize bipyrazole via acetylation, carbamoylation, bromination, dihydrochloride and dinitrate salt. The preparation of 3,4-dicarbonylated furan was further subjected to formylation reactions, and subsequent condensation reactions with hydrazine afforded bipyrazole. For example, the synthesis of furan, thiophene and pyrrole-3,4-dicarboxylic esters was reported, which can allow one to access an intermediate **3** for bipyrazole synthesis [10].

This procedure started with butyronitril undergoing a Claisen condensation reaction to afford sodium or potassium enolates **4**, which were then treated with SOCl_2_ to afford 3,4-dicyano-2,5-furandicarboxylate **5**. This compound was further subjected to decarboxylation reaction conditions with copper powder and heated to 160 °C, yielding 61–63% of the product (**6**) [12]. Using this dicyanofuran (**6**), Weis pursued reduction using DIBAL-H to access the dialdehyde compound (**3**). The author found that the compound is stable under acidic conditions; however, it was converted to the ring-opened form under basic conditions, resulting in the formation of the anionic form of tetraaldehyde. This procedure represents a classic synthesis of bipyrazole using a condensation reaction, but Domasevitch and co-workers reported that the procedure was not amenable for the synthesis of bipyrazole [13]. The alternative synthetic approach was started from a dialdehyde, acetylene and formaldehyde to generate an intermediate bis-trimethinium salt (Figure 1) [14]. The synthesis consists of a similar approach as the above-mentioned pathway: a formylation reaction to access tetraaldehyde compounds, which are masked as enamine and iminium functionalities. This sequence requires the use of phosgene and additional steps to finally synthesize bipyrazole.

Despite the high application potential of symmetric nitrogen-containing compounds in materials chemistry and coordination chemistry as tunable ligands, the synthetic procedures have been limited to traditional condensation reactions with low atom economy and redox efficiency. Not only are double condensation reactions difficult to control, but the lack of 1,3-dicarbonyl compounds has also hampered the development of facile preparation of pyrazole-based compounds. Therefore, many catalytic systems based on transition-metal catalysts have been investigated owing to the development of cross-coupling reactions, which will be discussed in this review.

## 3. Recent Advancement of Bipyrazole Syntheses

### 3.1. Transition Metal-Catalyzed Homocoupling of Azoles

Transition metal catalysts play an important role in the functionalization of C–H bonds of aromatic compounds and heteroaromatics and in the formation of new C–C bonds in the synthesis of homocoupled biaryl compounds [3]. The development in homocoupling reactions of pyrazoles, imidazoles and triazoles showed a broad range of catalytic activities in transition metals, including Cu, Fe, Ni, Pd, Rh and Ru, which were summarized in a review in 2012 [15]. This review will therefore summarize more recent examples (2010–2020) of homocoupling reactions of pyrazoles and azoles (imidazoles, triazoles, tetrazoles) and related compounds.

#### 3.1.1. Cu-Catalyzed Homocoupling of Various Azoles

The (oxidative) Glaser–Hay coupling reaction utilizes Cu(I) species as catalysts in the presence of a base under air to afford homo-coupled alkyne products [16]. An adaptation of the Glaser–Hay coupling reaction conditions to *N*-substituted 1,2,4-triazoles was realized starting from easily accessible triazole compounds. In 2009, Do and Daugulis developed Cu(II)-catalyzed homocoupling reactions of imidazoles using molecular oxygen as a terminal oxidant (Scheme 3) [17]. The method developed proved useful in the coupling of various heterocycles, including examples of a *N*-butyl-substituted imidazole and triazole, providing the desired products in up to 73% yield. The mechanism of the reaction was suggested to proceed via deprotonation of heteroarenes (pK_a_ < 35–37 for C2-H), followed by the complexation with metal additives, such as Li^+^. The *trans*-metalation step with the copper catalyst and subsequent oxidation yields the intermolecular homocoupling products between two arenes (Scheme 3).

Li et al. reported Cu(II)-mediated homocoupling reactions using a similar system at elevated temperatures (Scheme 4) [18]. This method utilizes an efficient and convenient approach to form C2-C2 bonds via a Cu(OAc)_2_ and O_2_-mediated oxidative intermolecular homocoupling reaction between two azoles. Once again, the oxidative coupling could be used to achieve high yields of homocoupled products.

#### 3.1.2. Pd-Catalyzed Homocoupling Reactions of Pyrazoles and Triazoles

Pd-catalyzed cross-coupling reactions are currently classified as some of the most flexible and valuable tools in organic synthesis [19,20]. A common observation is that Pd-catalyzed coupling reactions provide low but significant amounts of homocoupled by-products during cross-coupling reactions [21]. This is likely because the main focus of Pd-catalyzed coupling reactions is to form bonds between two different coupling partners, enabling highly productive homocoupling reactions. A handful of papers [22,23] have used the tactical placement of heteroatoms for the coordination of metals in the activation of specific C–H bonds. Other papers [24,25] utilize the possibility of the oxidative addition of Pd into a carbon halogen bond to form di-azole Pd complexes, which can undergo reductive elimination to form desired intermolecular homocoupled products. 

In 2012, the unexpected discovery of the intermolecular homocoupling between two pyrazoles was made by Salanouve et al. (Scheme 5) [22]. In an effort to study the reaction products obtained under different Suzuki–Miyaura conditions, this group discovered the formation of a C3–C3 intermolecular homocoupled product, resulting from C–H activation. The discovered bipyrazole compound was found to originate from a bis(3-ethoxy-1-(pyridin-2-yl)-1*H*-pyrazol-5-yl) palladium complex. Through oxidation with NCS, this species forms a Pd^IV^ complex, which then allows for reductive elimination of the homocoupled product in a yield of up to 34 % (Scheme 5b). 

Afanas’ev et al. reported a method for the homocoupling of 4-bromo-1,2,3-triazoles to access the corresponding 4,4′-bistriazoles in quantitative yields [24]. The intermolecular homocoupling was achieved by mixing 4-brominated triazoles with 1.0 mol% Pd(OAc)_2_, in the presence of additives (pinB)_2_, SPhos and KOH. This method is practical due to the use of readily available starting compounds and a simple experimental procedure, without inert atmosphere or solvent. The reaction possibly occurs through the formation of a 4-Bpin-substituted 1,2,3-triazole. Then, in a Suzuki–Miyaura fashion, the homocoupling of the boronic acid can occur (Scheme 6b).

A palladium-catalyzed, pyrazole-directed C(sp^2^)−H functionalization of the *N*-phenyl ring in *N*-phenylpyrazoles to afford a biaryl bis-pyrazole (via dehydrogenative homocoupling) was reported by Batchu et al. [23]. Multiple factors were shown to have significant influences on the outcome of the reaction. For example, it was discovered that if the reaction was performed under ca. 10 times dilution, then no homocoupling product was observed, and *N*-(*o*-hydroxyphenyl)pyrazoles were the major or the sole products. The reaction mechanism is suggested in Scheme 7b. After the first C–H activation, the Pd^II^-center is oxidized to Pd^IV^, which undergoes a second C–H activation followed by a reductive elimination to yield the homocoupled bipyrazole.

The homocoupling reaction between 1,2,3-triazole *N*-oxides by C–H activation at the C4 position was explored by Zhu et al. (Scheme 8) [26]. The reaction was designed to couple 1,2,3-triazole *N*-oxides with sodium arenesulfinates via palladium-catalyzed desulfitative cross-coupling, but it showed the possibility of homocoupling between 1,2,3-triazole *N*-oxides. Under unoptimized reaction conditions, the homocoupled *N*-oxide bistriazoles were generated with yields of up to 22%. The reaction likely occurs via deprotonation followed by exchange of OAc^−^ for the deprotonated 1,2,3-triazole *N*-oxides. The di-triazole *N*-oxide Pd^II^ complex then undergoes reductive elimination to yield the homocoupled bistriazole *N*-oxide product and Pd^0^, which can be oxidized by Ag_2_CO_3_.

Later, the full scope of the homocoupling reaction between pyrazole-*N*-oxides was investigated by Peng et al. (Scheme 9) [27]. The method showed high regioselectivity towards C–H homocoupling of 1,2,3-triazole *N*-oxides, using a Pd^II^ catalyst (5 mol%), Ag_2_CO_3_ and 1,10-phenanthroline under air. Following their previous work, it was found that the addition of the base *t*-BuOK and a catalytic amount of 1,10-phenanthroline allowed for yields of up to 93%.

The homocoupling reaction between trisubstituted 4-iodo pyrazoles in the C4 position was investigated by Jansa et al. (Scheme 10) [25]. The pyrazoles contain two to three pyridinyl substituents, which are synthesized from the reaction of 1,3-dipyridinyl-1,3-propanediones with 2-hydrazinopyridine or phenylhydrazine, affording the corresponding 1,3,5-trisubstituted pyrazoles. Iodination at the 4-position of the pyrazoles was achieved by treatment with I_2_/HIO_3_. The coupling of two pyrazole rings was achieved under Negishi cross-coupling reaction conditions by employing organozinc halides in combination with a Pd^II^ catalyst.

A more recent example of intermolecular homocoupling between two pyrazoles was discovered by Mercedes et al. (Scheme 11): through the palladium-catalyzed homocoupling of pyrazole boronic esters in the presence of air and water, 4,4-bipyrazole (H_2_bpz) and other symmetric bipyrazoles could be achieved [28]. The mechanism of the reaction is likely through a Suzuki–Miyaura-type cross-coupling between the boronic acids (Scheme 11b), with O_2_ being the oxidant for the Pd^0^ formed during the reductive elimination. 

#### 3.1.3. Ru-Catalyzed Homocoupling Reactions

Ackermann et al. demonstrated the chemoselective ruthenium-catalyzed C(sp^2^)–H bond arylations on triazol-4-yl and pyrazol-2-yl substituted arenes (Scheme 12) [29]. The authors discovered that *ortho*-substituted arenes favored oxidative intermolecular homocoupling reactions to afford dimerized products in high (or moderate) yields. The suggested reaction mechanism follows a double C–H bond activation with carboxylate assistance, followed by reductive elimination. 

#### 3.1.4. Ni-Catalyzed Reactions

Nickel is an Earth-abundant alternative to Pd and can access various reaction mechanisms, giving a distinct reactivity due to the facile oxidative addition reaction and readily accessible multiple oxidation states [30,31].

In 2000, Fanni et al. reported a new synthetic procedure for the efficient preparation of binuclear Ru(II) polypyridyl complexes, by performing intermolecular homocoupling between two 1,2,4-triazole bromide in the C3 position (Scheme 13) [32]. The halide functional group is necessary to mediate the Ni-catalyzed reaction in the presence of triphenylphosphine and a zinc reducing reagent.

The homocoupling between 4-bromotriazoles has been reported by Afanas´ev et al. using in situ generated Ni^0^ as a catalyst (Scheme 14) [24]. The homocoupling functions via an in situ formation of Ni^0^(PPh_3_)_4_ from NiCl_2_ and Zn^0^, which undergoes oxidative addition of two equivalents of 4-bromotriazoles, followed by a reductive elimination to form the desired homocoupled products (Scheme 14b).

#### 3.1.5. Rh-Catalyzed Homocoupling Reaction

Rhodium-catalyzed reactions have shown fruitful successes in organic reactions [33], particularly in C–H activation and C–C bond formation reactions [34,35]. heterocycle formation reactions [36,37], C–X bond formation reactions [38] and asymmetric synthesis [39].

Yue et al. explored the homocoupling of pyrazole silanes using Rh^I^ as a catalyst [40]. A suggested reaction mechanism includes a double *trans*-metalation reaction between vinylsilane and a rhodium intermediate in a single catalytic cycle (Scheme 15). While investigating an oxidant, the authors found out hexachloroacetone afforded higher yields of the desired product compared to conventional oxidants such as Cu^II^, TEMPO and chloranil.

#### 3.1.6. Fe-Catalyzed Homocoupling Reactions

Iron is one of the most Earth-abundant metals, and many iron complexes are commercially available. Surprisingly, despite its many advantages, iron has been relatively underrepresented in the catalysis of organic compounds until recently, compared to other transition metals.

In 2010, Luque et al. achieved the homocoupling of two pyrazole boronic acids utilizing nano-ferrite glutathione in a microwave (MW)-assisted aqueous reaction, providing an alternative to the palladium-catalyzed Suzuki coupling reaction (Scheme 16) [41].

### 3.2. Metal-Free Homocouplings of Azoles

In addition to the many transition metal-catalyzed routes described above, azole substrates have also been shown to undergo homocoupling reactions in the absence of a metal catalyst. Hypervalent iodine reagents can induce aryl–aryl bond formation in a wide variety of substrates, including pyrroles, pyrazoles, imidazoles and triazoles, as has been recently reviewed [42]. Other approaches to metal-free couplings rely on oxidants or photocatalysis to enact radical-mediated dimerizations. Some recent examples are presented below.

There is precedent for the metal-free homocoupling of pyrazolones through the use of oxidants like phenoxy radicals [43] or O_2_ [44]. A recent example extends this approach to a switchable aerobic oxidation, in which a pyrazol-5-one starting material undergoes either hydroxylation or a homocoupling, using O_2_ as an oxidant (Scheme 17a) [45]. Simple changes in reaction conditions control the outcome of this reaction; oxidation in dioxane in the presence of base favors hydroxylation, whereas a base-free reaction in acetonitrile favors the homocoupling. The homocoupling reaction was successful for 19 different substrates containing various substituents at positions R^1^ and R^2^, in yields up to 77% (Scheme 17b). Most reactions led to a mixture of diastereomers, with diastereomeric ratios ranging from 3.1:1 to 5.7:1, although three substrates led to a single diastereomeric product. The authors hypothesize that the homocoupling reaction proceeds through a radical intermediate, although mechanistic studies were not undertaken for this reaction.

Another homocoupling reaction was recently shown to yield a structurally similar bipyrazolone, but through remarkably different conditions: By refluxing a substituted pyrazole in excess thionyl chloride, the bipyrazolone was produced in high yields (Scheme 18a) [46]. This reaction was successful for nine substrates in yields from 62–89%. (The authors do not provide diastereomeric ratios of meso compounds and racemate for each substrate, but they report that dimeric pyrazolone (R^1^ = Ph) appeared to be a racemic mixture by ^1^H-NMR spectroscopy and X-ray crystallography). The reaction was shown to proceed under an atmosphere of N_2_ and under air, ruling out oxidation by air as the mechanism, and the authors propose a hypothetical mechanism via a di(pyrazolyl) sulfite intermediate (Scheme 18b).

Two recent reports describe the oxidative homocoupling of 4-amino-1,2,4-triazoles, yielding the corresponding symmetrical bis(triazoles). Yang and coworkers used the oxidant α-hydroxy-α-methylthioacetophenone to accomplish the homocoupling of five substituted 4-amino-1,2,4-triazoles in yields of 59–73% (Scheme 19a) [47]. Likewise, Zhao and coworkers accomplished the homocoupling of the unsubstituted 4-amino-1,2,4-triazole in 90% yield using sodium dichloroisocyanurate (SDCI) as the oxidant (Scheme 19b) [48].

Photocatalysis represents an alternative metal-free approach to pyrazole homocouplings. An early report of the photochemical oxidation of pyrazolidinones led to a homocoupled product [49]. More recently, Meng and coworkers used photocatalysis to accomplish the homocoupling of a bromopyrazolone [50]. By irradiating a solution of 4-bromo-3-methyl-1-phenyl-4,5-dihydropyrazol-5-one (PyHBr) and naphthalene in acetonitrile with a 300 W high-pressure mercury lamp, the authors obtained the homocoupled product in 64% yield (Scheme 20a). Interestingly, the authors discovered that by substituting phenathrene for naphthalene the dehydrogenated product was favored over the hydrogenated product, and the use of other arenes such as anthracene, benzene, acenaphthylene or indole did not lead to homocoupling at all. As the reaction relies on certain arenes and does not proceed under UV irradiation alone, the authors ruled out the homolytic bond cleavage of the carbon–bromine bond as the mechanism and instead proposed the mechanism shown in Scheme 20b.

Finally, a recent report of the decarboxylative fluorination of various heterocycles using Selectfluor contained the unexpected observation of homocoupling under certain conditions [51]. Specifically, substituted pyrazole-5-carboxylic acids and indole-2-carboxylic acids underwent homocoupling along with decarboxylative fluorination, yielding the corresponding fluorinated dimers (Scheme 21a). The homocoupling occurred in high yields (70–82%) for the pyrazole-based substrates and in moderate yields (38–58%) for the indole-based substrates (Scheme 21b). In the case of indole-2-carboxylic acid, the biproducts 2-fluoroindole and oxindole were generated along with the dimer (Scheme 21c). Mechanistic investigations indicated that neither of these biproducts is an intermediate in the dimer formation process, but the mechanism of the homocoupling was not elucidated. 

### 3.3. Applications in MOFs

The pyrazole functional group can coordinate with a transition metal as a ligand through the lone pair on the non-protonated nitrogen atom (Figure 2a) [52,53]. When the other nitrogen atom in the ring is deprotonated (to yield the pyrazolate), both nitrogen atoms can coordinate with metals, allowing pyrazolate to bridge metal centers and give rise to clusters, chains or coordination polymers (Figure 2b) [54,55]. Bipyrazolate ligands can coordinate up to four different metal centers (Figure 2c), making possible the construction of extended three-dimensional structures known as metal–organic frameworks (MOFs) [56]. Due to their crystallinity and internal porosity, metal–organic frameworks have shown promise in diverse applications, including gas separations [57], sensing [58], conductivity [59], drug delivery [60] and energy storage [61]. Numerous metal–organic framework structures have been designed and synthesized using poly(pyrazole) ligands as building blocks, as was reviewed by Galli and coworkers in 2016 [62]. Here we give an overview of recent studies on pyrazole-based metal–organic frameworks that have been published since the 2016 review, with an emphasis on 4,4′-bipyrazole (H_2_bpz) and its derivatives.

The unsubstituted H_2_bpz ligand has recently been used to form two new metal–organic frameworks (Figure 3). Volkmer and coworkers reacted H_2_bdp with a mixed-valent iron precursor to yield a nonporous metal–organic framework with Fe(II) ions as the framework nodes (CFA-10-as), and the authors were then able to convert the framework to a porous form upon thermal treatment (CFA-10) [63]. This single-crystal-to-single-crystal transformation was shown to result from the thermal decomposition of formate ions in the pores of CFA-10-as, resulting in a final porous structure that could not be achieved directly through solvothermal synthesis. A subset of these authors also described the synthesis of another H_2_bpz-based metal–organic framework, this time with Mn(III) ions as the framework nodes [64]. The structure of this framework, named Mn-CFA-6, was shown to contain two crystallographically and structurally distinct manganese centers, and the framework exhibited structural flexibility in response to the removal of guest molecules. Recently, Lee and coworkers observed a similar flexibility in the H_2_bpz-based framework Co(bpz), which undergoes reversible structural changes in response to gas adsorption or temperature change [28].

The tetramethylated bipyrazole ligand, 3,3′,5,5′-tetramethyl-4,4′-bipyrazole (H_2_Me_4_bpz), is another popular building block for metal-organic frameworks. In 2015, Sun and coworkers used H_2_Me_4_bpz to construct a series of Cu(I)-based coordination networks, including two novel metal–organic frameworks (Figure 4) [65]. These two frameworks contain halide and cyanide anions that coordinate with the Cu(I) nodes, along with the Me_4_bpz^2−^ ligands. The authors investigated these materials as heterogeneous catalysts for the azide–alkyne click reaction, because of the high density of Cu(I) sites present in the frameworks, by reacting benzyl azide with phenylacetylene in the presence of 1 mol % of a given metal–organic framework. The frameworks proved to be very effective catalysts for the formation of the substituted triazole, with the framework [Cu_6_(Me_4_bpz)_6_(CH_3_CN)_3_(CN)_3_Br]•2OH•14CH_3_CN leading to a 99% yield within 2 h. The authors ascribed this excellent catalytic performance to the framework’s high porosity and large pore diameter, which allow the Cu(I) active sites to be accessible to the reactants. The substrate scope for compound [Cu_6_(Me_4_bpz)_6_(CH_3_CN)_3_(CN)_3_Br]•2OH•14CH_3_CN was shown to include a variety of terminal azides and alkynes. This work was the first investigation of Cu(I)-based metal–organic frameworks as catalysts for click reactions, and the results indicate that this class of materials holds great promise in heterogeneous azide–alkyne catalysis.

Chen and coworkers attempted to increase the hydrophobicity of the hallmark metal–organic framework MOF-5 by substituting the traditional 1,4-benzenedicarboxylic acid (H_2_bdc) ligand with combinations of H_2_Me_4_bpz, naphthalene-1,4-dicarboxylic acid (H_2_ndc) and biphenyl-4,4′-dicarboxylic acid (H_2_bpdc) [66]. The authors synthesized one known framework (MAF-X10) and two new frameworks (MAF-X12 and MAF-X13) from these building blocks (Figure 5). The three metal–organic frameworks were investigated for fluorocarbon adsorption, using CHClF_2_ as a representative adsorbate, and all three were shown to possess high adsorption capacities for CHClF_2_. Grand canonical Monte Carlo simulations indicated that the strongest CHClF_2_ binding site in each framework was located in a hydrophobic pocket formed by two methyl groups from Me_4_bpz^2−^ ligands and the aromatic face of an adjacent phenyl ring, bearing out the authors’ hypothesis that increased ligand hydrophobicity would be beneficial for fluorocarbon adsorption. The compound MAF-X13 was shown to have the largest saturation uptake and the largest working capacity for CHClF_2_ adsorption, due to its type-IV isotherm shape.

Recently, Cao and workers took advantage of the differences in dative bond strength between carboylate ligands and azolate ligands to partially deconstruct mixed-ligand metal–organic frameworks [67]. The authors synthesized a metal–organic framework from a Zn(II) source and a mixture of H_2_Me_4_bpz and 1,4-benzenedicarboxylate ligands, and then used a pH 13 ammonia solution to dissolve the Zn–carboxylate bonds but not the Zn–pyrazolate bonds. This treatment led to a delamination of the three-dimensional framework, yielding two-dimensional sheets composed of Zn(II) nodes linked by Me_4_bpz_2_^−^ ligands. The authors also applied this process to an analogous Zn–triazolate–thiophenedicarboxylate MOF and used the resulting 2D Zn–triazolate sheets as building blocks to construct a library of new 3D frameworks, by combining the 2D sheets with various dicarboxylate linkers (Figure 6). This creative approach to metal–organic framework synthesis highlights the advantages of the strong metal–ligand bonding inherent to bipyrazole ligands like H_2_Me_4_bpz.

The sheer number of other recent examples of H_2_Me_4_bpz-based metal–organic frameworks emphasizes the utility of this bipyrazole ligand. These frameworks combine H_2_Me_4_bpz with various transition metals, including manganese [68], iron [69], cobalt [68,70,71], nickel [68], copper [72] zinc [73], silver [74,75,76] and cadmium [68,77,78], yielding a diverse array of framework topologies and functionalities. These frameworks were investigated for applications ranging from antibacterial therapy [76] to the photocatalytic degradation of organic pollutants [71], and chemists are sure to push the boundaries of H_2_Me_4_bpz-based metal–organic frameworks further in the coming years.

The dimethylated analogue of H_2_bpz is used in metal–organic framework synthesis less frequently than the tetramethylated analogue, but two recent studies show that H_2_Me_2_bpz can be used to form a variety of metal–organic frameworks. In 2017, Galli and workers reported two new metal–organic frameworks based on the reaction of H_2_bpz with a cobalt source and with a zinc source, respectively (Figure 7a). [79] The authors compared these two H_2_Me_2_bpz frameworks with the isostructural frameworks M(bpz) (M = Co, Zn) through techniques including gas adsorption and *ab initio* simulation, revealing that the methylated frameworks bound guest CO_2_ molecules more strongly than the nonmethylated frameworks but accommodated fewer overall guests because of the increased steric bulk of the H_2_Me_2_bpz ligand, relative to H_2_bpz. Additionally, in 2018 a subset of these authors reported a 3D polymeric network composed of Hg(II) ions linked by Me_2_bpz_2_^−^ ligands (Figure 7b). [80]

Finally, Galli and coworkers recently demonstrated the power of introducing functional groups onto the H_2_bpz scaffold, functionalizing this ligand with either NO_2_ or NH_2_ groups (Figure 8a) [81]. The authors incorporated these H_2_bpz derivatives into a series of Zn(II)-based metal–organic frameworks, each containing a different mixture of ligands. The authors found that frameworks containing a mixture of H_2_bpz and H_2_bpzNH_2_ ligands had the best gas adsorption properties in the context of N_2_/CO_2_ separations. The following year, Rossin and coworkers used the same three ligands to construct a series of Co(II)-based metal–organic frameworks (Figure 8b) [82]. The authors found that while the unfunctionalized Co(bpz) had higher O_2_ gas uptake relative to Co(bpzNO_2_) and Co(bpzNH_2_), ligand functionalization could tune the catalytic selectivity and performance of these materials when they were used for cumene oxidation and decomposition.

## 4. Conclusions

Despite the utility of bipyrazoles and other symmetric heterocycles as building blocks in coordination chemistry, traditional routes to these molecules were laborious, multi-step syntheses with poor atom economy and redox efficiency. In recent years, a number of improved routes to bipyrazoles have been published, employing an efficient homocoupling step that forms an aryl-aryl bond between the two halves of the product. Some of the most promising pyrazole homocoupling reactions rely on transition metal catalysts, especially palladium, as described above. We have also described other unique homocoupling strategies that proceed in the absence of metal catalysts, relying on UV irradiation, aerobic oxidation, or other diverse mechanisms. 

We anticipate that the coupling strategies provided in this review will allow metal–organic framework chemists to access a greater variety of bipyrazole-type ligands, beyond the small set of H_2_bpz derivatives described herein. As is obvious from the above studies, bipyrazole-based metal–organic frameworks are promising materials for a number of applications, especially gas adsorption and heterogeneous catalysis. The ability to diversify and functionalize bipyrazole ligands will afford chemists greater control over the resultant metal–organic framework properties, leading to even better adsorbents and catalysts.

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
