# Peer review of "Homocoupling Reactions of Azoles and Their Applications in Coordination Chemistry"

_molecules, 2020, doi:10.3390/molecules25245950_

Round 1

Reviewer 1 Report

In this manuscript "Homocoupling Reactions of Pyrazole-Type Compounds and Their Applications in Coordination Chemistry", the authors described recent advances in homocoupling reactions of nitrogenated heterocycles such as pyrazoles, imidazoles and triazoles and their application in material chemistry. After a summary of the traditional  syntheses of bipyrazoles, the authors presented recent bipyrazole syntheses based on metal catalysis. The review is organized by used metal and a special attention has been paid to mechanisms involved. After a last part on metal free synthesis of bipyrazoles, the authors finished the review by presenting their applications in metal-organic frameworks to show the utility of bipyrazoles.

This subject of bipyrazole-type compounds is of great importance and could reach a wide chemist audience that's why I believed that this review can be publishable after minor revisions. For a better understanding of the manuscript, I would have some suggestions:

  • page 2, scheme 2: number the compounds and bipyrazole should be added.
  • page 2, line 56-64: this paragraph should be rewritten for a better comprehension, especially by adding compound numbers.
  • page 3, line 70: copper powder.
  • page 3, line 71: Is it Trofimenko or Weis?
  • page 3, line 77: all the figure references of the manuscript are not found.
  • page 3, line 105: remove one "[13]".
  • page 5, scheme 4: "xylene"
  • page 6, line 142: scheme 5.
  • page 7, line 153: scheme 6.
  • page 8, line 160: remove "the".
  • page 8, line 163: scheme 7.
  • page 10, scheme 8: add the ratio of products.
  • page 12, line 208: scheme 11.
  • page 15, line 232: remove "and" at the end of the sentence.
  • page 15, line 242: scheme 14.
  • page 16, line 249-251: it should be rewritten.
  • page 16, line 254: scheme 15.
  • page 17, scheme 16: add the yield
  • page 20, line 294: why "racemic" is precise? Do you mean that the dr = 1:1?
  • page 23, scheme 21.c: add the yields.
  • references: the doi should be always written in the same way. The title of reference 8 should be in lower case letters.

Author Response

Thank you very much for detailed review from Reviewer 1. Here we addressed all the comments from the reviewer and we highlighted our answers in the attached file. Also, all the corrections are made in the manuscript with tracking option.

In this manuscript "Homocoupling Reactions of Pyrazole-Type Compounds and Their Applications in Coordination Chemistry", the authors described recent advances in homocoupling reactions of nitrogenated heterocycles such as pyrazoles, imidazoles and triazoles and their application in material chemistry. After a summary of the traditional  syntheses of bipyrazoles, the authors presented recent bipyrazole syntheses based on metal catalysis. The review is organized by used metal and a special attention has been paid to mechanisms involved. After a last part on metal free synthesis of bipyrazoles, the authors finished the review by presenting their applications in metal-organic frameworks to show the utility of bipyrazoles.

This subject of bipyrazole-type compounds is of great importance and could reach a wide chemist audience that's why I believed that this review can be publishable after minor revisions. For a better understanding of the manuscript, I would have some suggestions:

  • page 2, scheme 2: number the compounds and bipyrazole should be added.

Key compounds are numbered

  • page 2, line 56-64: this paragraph should be rewritten for a better comprehension, especially by adding compound numbers. The sentence is rewritten with numbering of compounds

page 3, line 70: copper powder. corrected

  • page 3, line 71: Is it Trofimenko or Weis? It’s Weis. Thank you for the correction
  • page 3, line 77: all the figure references of the manuscript are not found. The cross-reference is corrected
  • page 3, line 105: remove one "[13]". removed
  • page 5, scheme 4: "xylene" The scheme is corrected

page 6, line 142: scheme 5.  corrected

page 7, line 153: scheme 6. corrected

  • page 8, line 160: remove "the". Removed
  • page 8, line 163: scheme 7. corrected
  • page 10, scheme 8: add the ratio of products. The ratio from the literature is added
  • page 12, line 208: scheme 11. corrected
  • page 15, line 232: remove "and" at the end of the sentence. removed
  • page 15, line 242: scheme 14. corrected
  • page 16, line 249-251: it should be rewritten.
  • The sentence has been rewritten with new references for Rh-catalyzed reactions
  • page 16, line 254: scheme 15. corrected
  • page 17, scheme 16: add the yield
  • The paper doesn’t show the yield. We added conversion and selectivity instead.
  • page 20, line 294: why "racemic" is precise? Do you mean that the dr = 1:1? The authors clarified only one case (R1=Ph) which showed racemic mixture by X-ray single crystal structures. The 1H NMR spectroscopy showed only “one-set” of signals, therefore, the authors concluded that there is no meso-compounds.
  • page 23, scheme 21.c: add the yields. added
  • references: the doi should be always written in the same way. The title of reference 8 should be in lower case letters. Thank you for noticing this. The MDPI formatting will correct this during the editing stage.

Reviewer 2 Report

In this review article entitled “Homocoupling Reactions of Pyrazole-Type Compounds and Their Applications in Coordination Chemistry”, Ji-Woong Lee and co-workers gave an overview of all the recent protocols for the homo-coupling reactions of pyrazole and other five-membered N-heterocycles. In particular, the paper is organized in different sections: after a short introduction on the relevance of pyrazole and bipyrazole-based compounds, the first part is dedicated to the traditional approaches for the synthesis of bipyrazoles. In the second part, authors take into account all recent literature on the transition metal promoted homo-coupling reactions of pyrazole and other five-membered N-heterocycles: copper, palladium, ruthenium, nickel, rhodium and iron. A third part of the review was instead dedicated to all the metal-free approaches, including oxidative homocoupling and photocatalysts-based protocols. In the final part, authors shortly focused the attention on the application of bipyrazole and related compounds for the preparation of metal-organic-frameworks. Although the overall value of the present review appears quite interesting, unfortunately at this stage it is not yet suitable for publication on MDPI Molecules journal, thus suggesting a series of revisions.

1. At various points in the paper (starting from the title), authors used the term "pyrazole-type". Actually, I am not sure it is correct: in some point, authors described homocoupling reactions of triazoles, imidazoles, etc. In other words, authors tried to give an overview on the literature for all the five-membered N-heterocycles (at least, those bearing two or more nitrogen atoms), which are called azoles. Therefore, maybe it could be more correct, where possible, to change "pyrazole-type compounds" with "azoles".

2. On lines 56-81, the authors intention would be to show the traditional preparation methods of bipyrazoles. However, the attention was mainly focused on the synthesis of 3,4-dicarboaldehyde furan, rather than on bipyrazole compounds: this is also confirmed by Scheme 1. I believe that in this part authors should better emphasize (both in the text and the Scheme 1) how is possible to synthesize bipyrazoles from this furan.

3. On lines 91 and 95, authors specified that in the present review only the recent developments on homocoupling reactions of pyrazoles and other azoles will be considered, but they did not specify the time interval they will consider. Furthermore, I believe that it is fundamental to also cite some reviews focusing on the older papers on the same topic, in order to give to readers the possibility of a fully complete overview.

4. On lines 103-105 and 111-113, I believe that "(Scheme 3. a) Glaser-Hay type copper catalyzed homocoupling of various azoles resulting in homocoupling using molecular oxygen as a terminal oxidant. b) Substrate scope for the copper catalyzed homocoupling. [13])" should be simply changed with "(Scheme 3)".

5. On lines 127-128, authors wrote that Pd-catalyzed cross-coupling reactions are one of the most flexible and valuable tools in organic synthesis. I believe that some relevant, very recent reviews on the topic should be cited: for example, one for Suzuki-Miyaura reaction, one for acyl Sonogashira reaction, one for Negishi reaction, etc. In a very similar way, on lines 249-251 they wrote that Rhodium is an extremely useful metal in hydrogenation, C-H activation, allylic substitution, and numerous other metal-catalyzed reactions. In addition to change “numerous other metal-catalyzed reactions” with “numerous other reactions”, I suggest of citing recent representative reviews on the topic, for example: DOI: 10.1080/00397911.2018.1487070; DOI: 10.3390/catal10091012; etc.

6. Another important issue is the numbering of the schemes: on lines 121-122 a reference to the Scheme 4 is missing in the text; maybe for this reason, the numbering of all the other following schemes is wrong. In detail: on line 142, Scheme 4b should be actually Scheme 5b; on line 153, Scheme 5b should be actually Scheme 6b; on line 163, Scheme 6b should be actually Scheme 7b; on line 208, Scheme 10b should be actually Scheme 11b; on line 242, Scheme 13b should be actually Scheme 14b; on line 254, Scheme 14b should be actually Scheme 15b. Moreover, in the text I think that authors should refer also to all the Schemes depicted in the manuscript.

7. On line 294, authors indicated a compound as 3a. In Scheme 18, as well as in the text, compounds were not numbered. Therefore, either the numbering is removed for all compounds in the review, or it is inserted for all the compounds in this review. If authors decide not to assign numbers to the compounds, in the Scheme 18 letters a-i in the Table must be deleted.

8. Minor issues concerning the graphics. On Scheme 7, authors should change "a) Reaction conditions and mechanism" with "a) Reaction conditions", since the mechanism is depicted in the following point (b). On Scheme 19, authors should properly modify the scheme since -OCH3 and (68%) are partially overlapped.

9. Other minor issues. A) In several point of the manuscript I read the sentence “Error! Reference source not found.”, therefore I suggest authors to carefully check all the fields in the manuscript (these ones should be those of the Figures numbering). B) In many points of the manuscript, the letter N-, as well as the letter o- and the word ortho- should be written in italics: please, check carefully also this point. C) On line 198, I think that “by a Negishi cross-coupling” should be changed in “under Negishi conditions”. D) On line 205, “Through” should be changed with “through”. E) On line 227, “giving Ni-catalyzed reactions a distinct reactivity” should be changed in “giving a distinct reactivity”; similarly, on line 228, “nickel as a cross-coupling reagent.” should be changed in “nickel as a catalysts in homo- and cross-coupling reactions.”. F) On line 242, “scheme” should be changed with “Scheme”. G) On line 312, I think that “presents” should be changed with “represents”.

Author Response

We really appreciate for the critical comments from Reviewer 2. We addressed all the comments as shown in the revised manuscript and the point-by-point responses. 

Reviewer 2

In this review article entitled “Homocoupling Reactions of Pyrazole-Type Compounds and Their Applications in Coordination Chemistry”, Ji-Woong Lee and co-workers gave an overview of all the recent protocols for the homo-coupling reactions of pyrazole and other five-membered N-heterocycles. In particular, the paper is organized in different sections: after a short introduction on the relevance of pyrazole and bipyrazole-based compounds, the first part is dedicated to the traditional approaches for the synthesis of bipyrazoles. In the second part, authors take into account all recent literature on the transition metal promoted homo-coupling reactions of pyrazole and other five-membered N-heterocycles: copper, palladium, ruthenium, nickel, rhodium and iron. A third part of the review was instead dedicated to all the metal-free approaches, including oxidative homocoupling and photocatalysts-based protocols. In the final part, authors shortly focused the attention on the application of bipyrazole and related compounds for the preparation of metal-organic-frameworks. Although the overall value of the present review appears quite interesting, unfortunately at this stage it is not yet suitable for publication on MDPI Molecules journal, thus suggesting a series of revisions.

  1. At various points in the paper (starting from the title), authors used the term "pyrazole-type". Actually, I am not sure it is correct: in some point, authors described homocoupling reactions of triazoles, imidazoles, etc. In other words, authors tried to give an overview on the literature for all the five-membered N-heterocycles (at least, those bearing two or more nitrogen atoms), which are called azoles. Therefore, maybe it could be more correct, where possible, to change "pyrazole-type compounds" with "azoles".

This is highly accurate criticism. The intention of the review is to fit into the special issue – however, we realized that we should not limit the scope of the review to only pyrazoles, since similar heterocyclic compounds are going to find more applications in materials chemistry once the synthetic procedures have matured. Now we have corrected the abstract and subchapter 3 and replaced “pyrazole-type” with “azoles” where necessary.

  1. On lines 56-81, the authors intention would be to show the traditional preparation methods of bipyrazoles. However, the attention was mainly focused on the synthesis of 3,4-dicarboaldehyde furan, rather than on bipyrazole compounds: this is also confirmed by Scheme 1. I believe that in this part authors should better emphasize (both in the text and the Scheme 1) how is possible to synthesize bipyrazoles from this furan.

The intention was to illustrate how difficult it is to access bipyrazoles’ precursors. In particular, tetraaldehyde, which is the key intermediate, requires multiple steps of synthesis. The last step – bipyrazole formation with hydrazine – seems quite straightforward. Based on the reviewer’s comment, Scheme1 was corrected for the bipyrazole synthesis step via dehydration with hydrazine.

  1. On lines 91 and 95, authors specified that in the present review only the recent developments on homocoupling reactions of pyrazoles and other azoles will be considered, but they did not specify the time interval they will consider. Furthermore, I believe that it is fundamental to also cite some reviews focusing on the older papers on the same topic, in order to give to readers the possibility of a fully complete overview.

The timeline is specified and previous review articles were cited.

  1. On lines 103-105 and 111-113, I believe that "(Scheme 3. a) Glaser-Hay type copper catalyzed homocoupling of various azoles resulting in homocoupling using molecular oxygen as a terminal oxidant. b) Substrate scope for the copper catalyzed homocoupling. [13])" should be simply changed with "(Scheme 3)".

Corrected

  1. On lines 127-128, authors wrote that Pd-catalyzed cross-coupling reactions are one of the most flexible and valuable tools in organic synthesis. I believe that some relevant, very recent reviews on the topic should be cited: for example, one for Suzuki-Miyaura reaction, one for acyl Sonogashira reaction, one for Negishi reaction, etc. In a very similar way, on lines 249-251 they wrote that Rhodium is an extremely useful metal in hydrogenation, C-H activation, allylic substitution, and numerous other metal-catalyzed reactions. In addition to change “numerous other metal-catalyzed reactions” with “numerous other reactions”, I suggest of citing recent representative reviews on the topic, for example: DOI: 10.1080/00397911.2018.1487070; DOI: 10.3390/catal10091012; etc.

Suggested references and additional citations were added to support the text (Pd: 19-20, Rh: 31-37)

  1. Another important issue is the numbering of the schemes: on lines 121-122 a reference to the Scheme 4 is missing in the text; maybe for this reason, the numbering of all the other following schemes is wrong. In detail: on line 142, Scheme 4b should be actually Scheme 5b; on line 153, Scheme 5b should be actually Scheme 6b; on line 163, Scheme 6b should be actually Scheme 7b; on line 208, Scheme 10b should be actually Scheme 11b; on line 242, Scheme 13b should be actually Scheme 14b; on line 254, Scheme 14b should be actually Scheme 15b. Moreover, in the text I think that authors should refer also to all the Schemes depicted in the manuscript.

All the cross-reference tags are not fixed and scheme numbers were added in the main text.

  1. On line 294, authors indicated a compound as 3a. In Scheme 18, as well as in the text, compounds were not numbered. Therefore, either the numbering is removed for all compounds in the review, or it is inserted for all the compounds in this review. If authors decide not to assign numbers to the compounds, in the Scheme 18 letters a-i in the Table must be deleted.

The alphabetic letters in Scheme 18 has been deleted and the compound is described as the product with (R1 = Ph) for better readability.

  1. Minor issues concerning the graphics. On Scheme 7, authors should change "a) Reaction conditions and mechanism" with "a) Reaction conditions", since the mechanism is depicted in the following point (b). On Scheme 19, authors should properly modify the scheme since -OCH3 and (68%) are partially overlapped.

Scheme 7 and 19 have been corrected.

  1. Other minor issues. A) In several point of the manuscript I read the sentence “Error! Reference source not found.”, therefore I suggest authors to carefully check all the fields in the manuscript (these ones should be those of the Figures numbering).

I am terribly sorry for these unprofessional mistakes. All the cross-reference tags are not fixed.

B) In many points of the manuscript, the letter N-, as well as the letter o- and the word ortho- should be written in italics: please, check carefully also this point.

Corrected (for example, Line 155, 159, 169-176)

C) On line 198, I think that “by a Negishi cross-coupling” should be changed in “under Negishi conditions”.

corrected

D) On line 205, “Through” should be changed with “through”.

corrected

E) On line 227, “giving Ni-catalyzed reactions a distinct reactivity” should be changed in “giving a distinct reactivity”; similarly, on line 228, “nickel as a cross-coupling reagent.” should be changed in “nickel as a catalysts in homo- and cross-coupling reactions.”.

Corrected

F) On line 242, “scheme” should be changed with “Scheme”.

Corrected

G) On line 312, I think that “presents” should be changed with “represents”.

Corrected

Round 2

Reviewer 2 Report

In this revised version of the paper, authors addressed satisfactorily all the issues listed in my previous report, thus allowing to improve the quality of the work. I believe that the present review now meets the standards for publication in the MDPI Molecules journal, thus suggesting its acceptance in the present form.